# THE CONNECTION BETWEEN OUT-OF-DISTRIBUTION GENERALIZATION AND PRIVACY OF ML MODELS

## ABSTRACT

With the goal of generalizing to out-of-distribution (OOD) data, recent domain generalization methods aim to learn "stable" feature representations whose effect on the output remains invariant across domains. Given the theoretical connection between generalization and privacy, we ask whether better OOD generalization leads to better privacy for machine learning models, where privacy is measured through robustness to membership inference (MI) attacks. In general, we find that the relationship does not hold. Through extensive evaluation on a synthetic dataset and image datasets like MNIST, Fashion-MNIST, and Chest X-rays, we show that a lower OOD generalization gap does not imply better robustness to MI attacks. Instead, privacy benefits are based on the extent to which a model captures the stable features. A model that captures stable features is more robust to MI attacks than models that exhibit better OOD generalization but do not learn stable features. Further, for the same provable differential privacy guarantees, a model that learns stable features provides higher utility as compared to others. Our results offer the first extensive empirical study connecting stable features and privacy, and also have a takeaway for the domain generalization community; MI attack can be used as a complementary metric to measure model quality.

## 1 INTRODUCTION

Generalization of machine learning (ML) models to out-of-distribution data (domains) is important for the success of their deployment in practice. To address this challenge, several domain generalization (DG) learning techniques are proposed that achieve improved out-of-distribution (OOD) accuracy (Dou et al., 2019; Piratla et al., 2020; Asadi et al., 2019; Ilse et al., 2020). Among them, state-of-the-art solutions rely on the idea of learning *stable* feature representations whose effect on the output remains invariant across domains (Arjovsky et al., 2019; Mahajan et al., 2021). Since these stable features correspond to the *causal* mechanism by which real world data is generated, stable features-based learning methods reduce the OOD generalization gap (Peters et al., 2016).

Concurrently, there is a growing literature on ensuring privacy guarantees for ML models since privacy is an important requirement for deploying ML models, especially in sensitive domains. Privacy attacks such as *membership inference* can leak information about the data used to train models (Shokri et al., 2017). Existing defenses such as differentially-private training prevent the leakage but hurt the model accuracy significantly (Truex et al., 2018; Zanella-Béguelin et al., 2020; Carlini et al., 2019).

Theoretically, failure to generalize (i.e., overfitting) has been shown to be a sufficient condition for membership inference attacks when evaluated on the same distribution as training data (Yeom et al., 2018). When evaluated on a different distribution, as is often the case in real-world domain generalization tasks, the membership attack risk is expected to worsen for standard algorithms (e.g., empirical risk minimization) since generalization is harder. From the mitigation perspective under multiple data distributions, Tople et al. (2020) has theoretically shown that causal models that learn stable features are robust to membership inference attacks as compared to standard neural networks training. However, their result holds in the limited setting where all true stable features are known apriori (*causal sufficiency*), which is unrealistic for most applications.

With the empirical advances in DG algorithms that learn stable features, we ask whether better OOD generalization can provide a viable route to membership inference robustness. Specifically, we ask

two questions: Would state-of-the-art domain generalization techniques that provide better out-of-distribution generalization lead to membership privacy benefits? And is learning stable features necessary for membership privacy benefits? To answer these questions, we conduct an extensive empirical study using recent DG algorithms trained over multiple datasets: simulated datasets, semi-simulated DG benchmarks, and a real-world Xrays dataset.

Based on our experiments, we *do not* find a direct relationship between better OOD generalization and privacy. Our work provides the first empirical evidence that a higher generalization gap is *not necessary* to perform MI attacks in out-of-distribution deployment scenarios. Prior work has only confirmed the sufficiency argument from Yeom et al. (2018) for practical ML algorithms under in-distribution deployment setting (Shokri et al., 2017; Salem et al., 2019). Specifically, we find that an algorithm with lower OOD generalization gap may have worse membership privacy compared to another with higher generalization gap.

Instead, our results indicate that the ability of a model to learn stable features is a *consistent indicator* of its privacy guarantees in practice. This finding is true irrespective of whether the algorithm in theory relies on capturing stable features or not, thus distinguishing our findings from those of Tople et al. (2020). When an ideal stable feature learner can be constructed, as in our synthetic datasets, the resultant model has the best privacy across all our experiments (and also best OOD generalization), Moreover, with the same provable differential privacy guarantee, a stable feature learning algorithm obtains better accuracy than ERM, showing the privacy benefits of learning stable features.

For the DG literature, our results point to a viable metric for measuring stability of learnt features, a fundamentally difficult task. Rather than OOD generalization, we find that MI robustness is better correlated with amount of stable features learnt and can be used to evaluate quality of DG algorithms. To summarize, our contributions are:

- Through extensive experiments, we show that better out-of-distribution generalization does not always imply better privacy. We explain the result through a simple counter-example.
- We provide the first empirical evidence that models that learn stable features are robust to membership inference attacks irrespective of the learning objective, thereby extending the theoretical results from Tople et al. (2020). Further, for the same added noise for differential privacy, an algorithm with stable features provides better utility.
- Current DG methods aimed to learn stable features do not do so even when they exhibit good OOD generalization. Therefore, we propose MI attack metric to evaluate quality of the learnt features, since it measures stable features better than OOD accuracy.

## 2 BACKGROUND & PROBLEM STATEMENT

We investigate connections between two streams of work: machine learning algorithms for generalization to unseen distributions and membership privacy risks of machine learning models.

**Out-of-Distribution/Domain Generalization.** As standard training algorithms often fail at generalizing to data from a different distribution than the training data, there is increased attention on domain generalization algorithms that perform well on unseen distributions (Arjovsky et al., 2019; Ahuja et al., 2020; Peters et al., 2016; Hendrycks & Dietterich, 2019; Hendrycks et al., 2020; Piratla et al., 2020; Gulrajani & Lopez-Paz, 2020; Mahajan et al., 2021). In a typical domain generalization (DG) task, a learning algorithm is given access to data from multiple domains at the training time. The goal is to train a model that has high accuracy on data from both the training domains as well as new unseen domains that the model may encounter once it is deployed in the wild. Formally, different domains correspond to different feature distributions $\mathcal{P}(\mathcal{X})$ (*covariate shift*) and/or different conditional distributions $\mathcal{P}(\mathcal{Y}|\mathcal{X})$ (*concept drift*). To emulate realistic scenarios, the unseen domains are constrained in a reasonable way, for example, domains might be different locations or regions, different views or lighting conditions for photos, etc. Hence, prior work assumes that all domains share some stable features $\mathcal{X}_\mathcal{C}$ that *cause* the output label $\mathcal{Y}$, for which the ideal function $\mathcal{P}(\mathcal{Y}|\mathcal{X}_\mathcal{C})$ remains invariant across all the domains (Arjovsky et al., 2019).

**Stable Features or Stable Representation.** Given $(\mathbf{x}, y)_{i=1}^{|d_k|}$ data over $k$ training domains $\{d_1, d_2..d_k\}$, a learnt representation $\Phi(\mathbf{x})$ is called stable if the prediction mechanism conditioned on them, $P_d(y|\Phi(\mathbf{x})$ remains invariant for all the domains $d \in \{d_1, d_2, ..d_k\}$. Additionally, we aim to learn stable representations such that classifier learnt upon them is optimal for generalization to unseen domains. Therefore, it is not surprising that state-of-the-art DG algorithms are designed to

learn these stable or causal $\mathcal{X}_C$ features. One class of methods learn causal representation by aiming that any pair of inputs that share the same causal features have the same representation. They use matching-based regularizer such as MatchDG (Mahajan et al., 2021), or Perfect-Match that assumes knowledge of ground-truth matched pairs. Other methods like Invariant Risk Minimization (IRM) build a representation that is simultaneously optimal across all training domains (Arjovsky et al., 2019; Ahuja et al., 2020), a property that is satisfied by the causal features. Another recent approach (CSD) aims to separate out input features into two parts such that one of them has common feature weights across domains, and uses only those common (stable) features for prediction (Piratla et al., 2020). We use all these DG algorithms for our experiments (Perfect-Match, MatchDG, IRM, CSD), in addition to non-stable learning algorithms, Random-Match (that matches same-class inputs from different domains (Motiian et al., 2017)) and ERM. Details of these algorithms are in Supp. B.1.

**Membership Privacy.** Ensuring membership privacy of ML models means hiding the knowledge of whether a particular record (or user providing the record) participated in the training of the model. Leakage of such membership information poses a serious privacy concern in sensitive applications such as healthcare. Remarkably, with only black-box access to the ML model (and no access to the training data or the model parameters), attacks have been created that can guess the membership of an input with high accuracy (Shokri et al., 2017). Overfitting to training dataset has been shown as one of the main reasons for membership leakage, both theoretically and empirically (Yeom et al., 2018; Shokri et al., 2017). Under bad generalization properties, the confidence in the prediction values and the resultant accuracy will be lower on a non-training point than a training point, and this difference has been exploited to design different variants of the membership inference attack (Salem et al., 2019; Song & Mittal, 2020; Nasr et al., 2019). For mitigating these attacks, training using differential privacy is the most widely used approach although it incurs substantial degradation in model utility (Dwork et al., 2014; Abadi et al., 2016).

**Research Questions.** While prior work has studied the relationship of generalization and membership privacy of ML models for in-distribution setting, it is not well-established how this relationship transfers when we consider out-of-distribution evaluation data. This forms our first research question *RQ1. Does better out-of-distribution generalization always lead to better membership privacy?* As we shall see, the answer to the above question is negative. Given recent theoretical work (Tople et al., 2020) that shows models learning stable features are more differentially private and robust to membership privacy under certain conditions (such as causal sufficiency), we ask a more nuanced question: *RQ2. Do methods that learn more stable features achieve better membership privacy?*

**Evaluation Metrics.** To answer these questions, we compare DG models on three metrics, a) **Utility:** Out-of-Distribution (OOD) accuracy on unseen domains; b) **Privacy:** attack accuracy using the loss-based membership inference attack (Yeom et al., 2018); and c) **Stable Features:** The amount of stable features learnt. The first is the standard metric used to evaluate DG algorithms (Gulrajani & Lopez-Paz, 2020) and the second is the simplest membership privacy attack metric that distinguishes members (train domains) from non-members (test domains) with only access to the loss values. The third, measuring stability of learnt features, is a fundamentally difficult question for DG models since stable features may correspond to unobserved high-level latent features. Therefore, we only measure it on the synthetic and semi-synthetic datasets where reliable metrics can be constructed. Further details regarding the evaluation metrics are provided in Supp. A.

## 3 EVALUATION RESULTS

We evaluate state-of-the-art domain generalization (DG) algorithms on image benchmarks like Rotated Digit and Fashion MNIST, and a real-world ChestXray dataset.

### 3.1 ROTATED DIGIT AND FASHION MNIST DATASETS

**Dataset Description.** The Rotated Digit dataset (Piratla et al., 2020) is built upon the MNIST handwritten digits, where we create multiple domains by rotating each digit with the following angles $0°, 15°, 30°, 45°, 60°, 75°, 90°$ (Ghifary et al., 2015). Angles from $15°$ to $75°$ constitute the training domains and the angles $0°$ and $90°$ are the unseen test domains. In the Rotated Fashion-MNIST dataset (Xiao et al., 2017), the digits are being replaced by fashion items. We sample 2000 data points from each of the training and test domains to create our training and test datasets similar to prior work (Piratla et al., 2020; Mahajan et al., 2021). For the validation set, we sample additional 400 data points from each training domain. We vary the number of training domains (5, 3 and

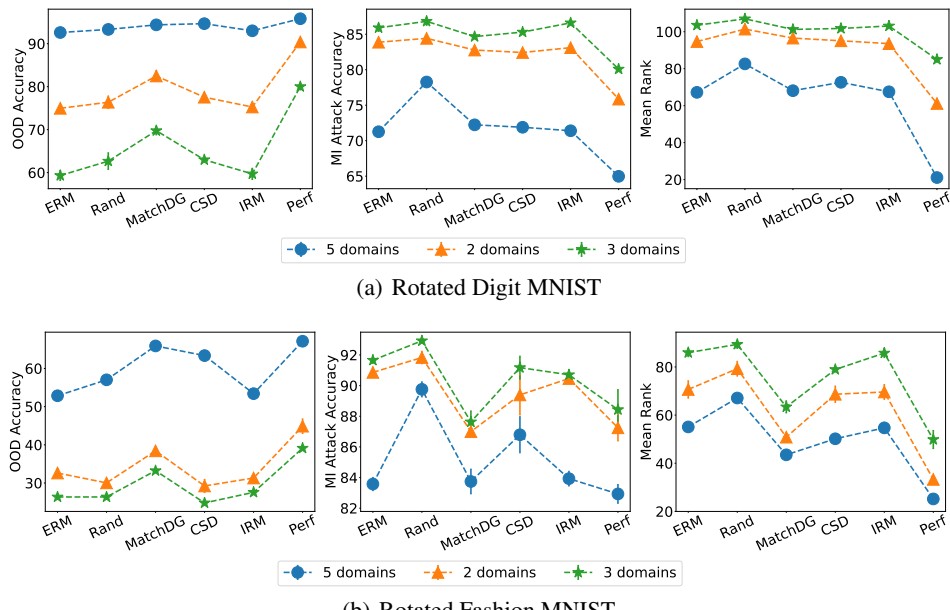

(a) Rotated Digit MNIST

(b) Rotated Fashion MNIST

Figure 1: OOD accuracy (left), Membership Inference attack accuracy (center) and Mean Rank metric (right) for the MNIST datasets. The line shapes denote different number of training domains. The error bars represent standard deviation over three runs.

2) to understand its effect on OOD accuracy and privacy. Further details on the dataset and our implementation are present in Supp. B.2

**Measuring Stable Features.** While the intuition is that an object's shape is the stable feature, it is not possible to measure it directly. Hence, we use the *mean rank* metric from Mahajan et al. (2021). Using a distance metric between any two inputs over the learnt representation, it measures the rank of distances for ground-truth pairs of objects that share the same base object (and hence same stable features). This metric is computable only because we know the ground-truth pairs by mapping back rotated images to their original image. A lower mean rank means more stable features are learnt by the model as explained in Supp. A.4.

**Results on RQ1 and RQ2.** Figure 1 shows the results for the Rotated Digit and Fashion MNIST datasets. We find that algorithms that aim to learn stable features do not learn them fully, so we compare the extent to which stable features are learnt across algorithms. For these datasets, we observe that all the methods perform equally well on the training dataset and hence the OOD accuracy (left panel of Figure 1 )—i.e., accuracy on the test domains—directly captures the out-of-distribution generalization gap. Higher OOD accuracy indicates a lower generalization gap. Comparing the OOD accuracy and MI attack accuracy in Figure 1 (center panel), we present our findings for both datasets together since the observations are similar. On the first research question (RQ1):

- On balance, DG methods that achieve higher OOD accuracy have lower risk of membership inference. For example, the Perfect-Match algorithm achieves the highest OOD accuracy and the lowest MI attack accuracy in all the different setting exhibiting its robustness to privacy leakage.
- However, the above phenomenon is *not true* always. Random-Match has better OOD accuracy than method such as IRM, but demonstrates higher susceptibility to MI attacks. Therefore, high OOD accuracy or a low generalization gap does not imply better membership privacy. To the best of our knowledge, this is the first result on a confirming Yeom et al. (2018)'s theoretical result of overfitting or high generalization gap is not necessary for MI attacks in practice.

To answer the second question (RQ2), we compare the MI attack accuracy in Figure 1 (middle panel) and the stable features or mean rank in Figure 1 (right panel).

- For both the datasets and across different (2, 3 and 5) training domains, the MI attack accuracy of the DG methods can be consistently explained by the amount of learnt stable features as reported by the Mean Rank metric. The Perfect-Match algorithm which has the lowest attack accuracy also always has the least mean rank, indicating the ability to learn the highest amount of stable features as compared to others.

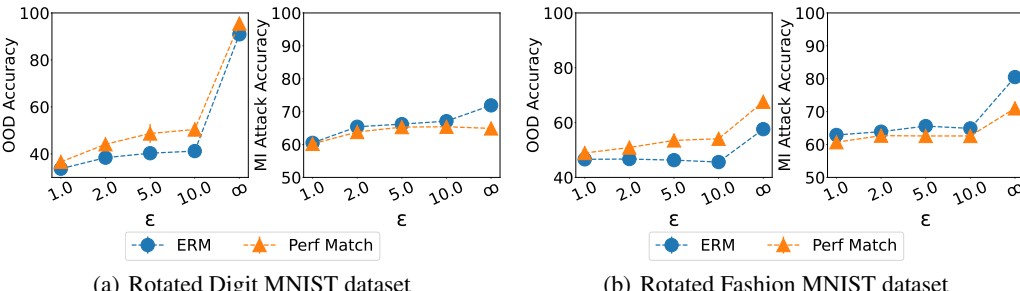

(a) Rotated Digit MNIST dataset  (b) Rotated Fashion MNIST dataset

Figure 2: OOD and MI attack accuracy of differentially-private ERM and Perfect-Match algorithms

- The stable feature metric (Mean Rank) can explain the negative result for RQ1. Random-Match has a higher OOD accuracy than ERM or IRM but has a higher mean rank, indicating that it does not learn stable features. Thus, we obtain a positive answer to RQ2: methods that learn more stable features provide better membership privacy empirically, on both datasets. Our empirical results apply to methods and datasets in the setting where true stable features are not known, and hence extend the purely theoretical results from Tople et al. (2020).

**MI as a metric for stable features.** Our experimental results also provide a metric for a critical evaluation problem in DG literature: measuring stability of learnt features. Comparing the left and right panels, better OOD generalization does not always imply better stability of features learnt. For instance, Random-Match algorithm obtains higher OOD accuracy than IRM or ERM, but has higher mean rank than these algorithms. In contrast, the MI attack metric (center panel) correlates strongly with the mean rank metric and produces the same ordering over algorithms. This indicates that the MI attack accuracy can be a useful metric to evaluate quality of the learnt representation by a DG algorithm on real-world datasets where it is difficult to measure the amount of learnt stable features.

**Comparison to Differential Privacy.** Lastly, results from Figure 1 and our affirmative response to RQ2 imply that stable-feature based learning methods such as Perfect-Match are capable of simultaneously providing better privacy and utility. Alternatively, *differentially-private* (DP) training which is the best known defense for MI attacks is infamous for degrading model utility. $\epsilon$-differentially-private mechanisms require addition of noise calibrated systematically to hide the contribution of a single data-point during training (Dwork et al., 2014; Abadi et al., 2016; Papernot et al., 2017; Hamm et al., 2016). The value of $\epsilon$ quantifies the amount of privacy guarantees with lower $\epsilon$ values providing stronger privacy. However, the addition of noise results in degradation of model utility proportional to the achieved privacy. This raises the question *if stable feature-based learning algorithms provide better empirical membership privacy as compared to a differentially-private model while preserving model utility.*

To answer this, we compare the results on DG training methods when trained with $\epsilon$-DP guarantees and without ($\epsilon = \infty$). Specifically, we compare ERM and Perfect-Match algorithms, as examples of a standard ML model and a stable feature model. We train these models on both the MNIST datasets using the Pytorch Opacus library v0.14 for $\epsilon$ values ranging from 1 to 10. Additional implementation details for the DP training are in Supp. B.2. Figure 2 presents the results for Rotated MNIST (a) and Fashion-MNIST (b) for differentially-private ERM and Perfect-Match. Our findings are:

- For the Digit MNIST dataset, the empirical membership privacy as measured via the attack accuracy of Perfect-Match at $\epsilon = \infty$ is comparable to that at $\epsilon = 5$ or 10, whereas for ERM the attack accuracy at $\epsilon = \infty$ is substantially worse than at any $\epsilon <= 10$. For Fashion-MNIST, the drop in attack accuracy from $\epsilon = \infty$ to 5 for Perfect-Match is less than that for ERM. This shows that Perfect-Match provides some inherent mitigation or robustness to MI attacks as compared to standard ERM method. Although, note that at $\epsilon = \infty$, there is no provable privacy guarantee.
- For the same $\epsilon$ guarantee, the utility of Perfect-Match is consistently higher than ERM, with a significant gap on Fashion-MNIST dataset for larger $\epsilon$ values. This shows that stable-feature based learning algorithms are a promising way to balance the privacy-utility trade-off.

## 3.2 CHESTXRAY DATASET

**Dataset.** To study a more practical scenario where stable features cannot be directly measured, we use the dataset of Chest X-ray images from Mahajan et al. (2021). It comprises of data from different

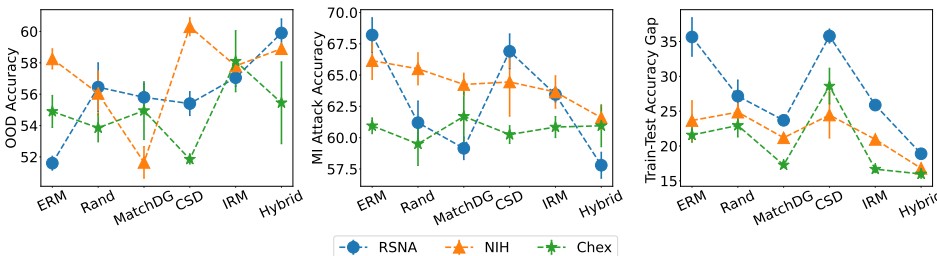

Figure 3: OOD accuracy (left), MI attack accuracy (middle) and Train-test accuracy gap (right) for ChestXRay dataset. The legend mentions the test domain.

hospitals: NIH (Wang et al., 2017), ChexPert (Irvin et al., 2019) and RSNA (rsn, 2018). The task is to train a classifier that predicts whether a patient suffers from Pneumonia or not. The dataset contains spurious correlation in the training domains due to a vertical translation that shifts all the data points with class 0 downwards. No such spurious correlation is present in the test domain.

**Metrics.** As there is no known base object for this dataset, Perfect-Match method cannot be implemented. Instead, we use the Hybrid method from Mahajan et al. (2021) that utilizes self-augmentations to create input pairs with the same stable features. Further, without the knowledge of perfect matches across domains (pairs with same stable features), we cannot compute the mean rank metric. Therefore, this dataset resembles the real-world scenario where it is not possible to reliably measure the feature stability. Instead, we report the generalization gap for easier comparison.

**Results.** Figure 3 shows the result for different DG algorithms when trained on the ChestXray dataset using two domains and evaluated on the third domain. We present our findings below:

- For RSNA as the test dataset, the MI attack accuracy can be explained by the generalization gap results for each of the training algorithms, with the Hybrid approach providing the best privacy followed by MatchDG. When evaluated on the NIH dataset, the MI attack accuracy and the generalization gap are in agreement again.
- For the ChexPert dataset, however, CSD has a higher generalization gap than MatchDG but has a similar MI attack accuracy (considering the standard deviations). Thus, while MNIST results showed that a low OOD generalization gap does not imply better MI privacy, here we find that higher OOD generalization gap does not mean worse MI privacy. We speculate that CSD has learnt stable features and that would explain the better MI attack accuracy results.

For *RQ1*, the above results confirm that OOD generalization does not have a direct relationship with privacy. For *RQ2*, as there is no reliable way to measure stable features, we are unable to conclude. As future work, we acknowledge the importance of designing metrics for stable features for real-world datasets.

## 4 EXPLANATIONS AND KEY RESULTS

From a theoretical standpoint, Yeom et al. (2018) show that overfitting is sufficient, but not necessary for MI attacks. Our study on state-of-the-art DG methods shows that the reality is much more complicated. Using OOD generalization as a measure of overfitting, we find no direct relationship between OOD generalization and MI privacy when comparing *between* algorithms: an algorithm with higher OOD generalization can have worse MI privacy compared to one with lower OOD generalization; and vice versa.

On the use of stable features for privacy, our findings extend the theoretical claim from Tople et al. (2020) to real-world empirical settings where the causal graph is not known and all causal parents may not be observed. We directly measure the stability of features learnt, unlike Tople et al. (2020) that assumed that a DG algorithm like IRM would learn stable features (an assumption that we've shown is not always true). We provide further explanations for our findings below.

### 4.1 GENERALIZATION GAP AND PRIVACY

From both MNIST and ChestXray experiments, we observe that generalization gap cannot always explain the robustness to membership attacks. We provide an example case study to shed more light

on the nature of relationship between OOD generalization and privacy: specifically, we show that OOD generalization for classification depends on the classification probability distribution while MI attack accuracy depends on the loss distribution for a classifier. These are correlated but inferences based on thresholds on these quantities need not be the same.

Consider a binary classification problem and two methods A and B such that they have similar loss distribution on the train dataset and thus obtain same performance on the train dataset. However, their loss distribution differs on the test dataset; say method B has better classification accuracy on the test dataset. Therefore, method B has smaller generalization gap than method A. But their loss distribution on the test dataset can still lead to method B being worse at defense against the MI attack than method A.

We explain this setting in Figure 1, shown only for data with class $y = 1$ (a similar argument works for $y = 0$). The threshold (purple horizontal line) for loss based MI attack is chosen as the mean loss of all train dataset points (*members*) with class 1. The orange vertical line shows the classification threshold. Since the loss distribution on train dataset is same for both the methods, they would obtain same error on prediction of true member data points. Note that there is a higher density of misclassified non-member data points for method A than method B as its performance is worse on the test dataset. But the non-members correctly classified by method A have loss distribution that overlaps with the loss distribution of members, which implies the attacker will incorrectly classify the overlapping non-members to be members. However, for method B, there is no overlap in the loss distribution of members and non-members, therefore the attacker can obtain perfect accuracy. Thus, method A provides better defense against MI attack than method B, despite having worse generalization gap. This shows that generalization gap cannot always explain the MI attack accuracy.

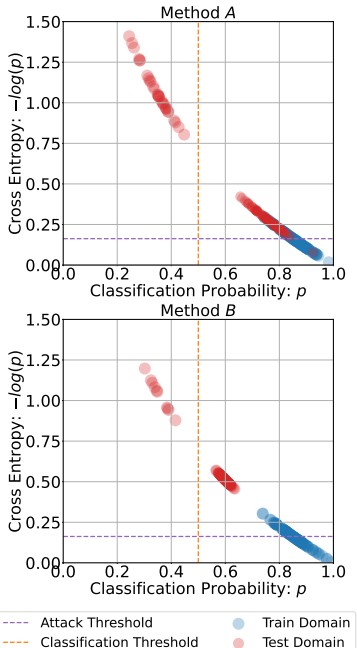

Figure 4: Loss distribution for the class y=1 for two method A and B

With a lower generalization gap but higher MI attack accuracy, method B emulates the behavior of Random-Match algorithm on MNIST datasets. Similarly, Method A emulates the CSD results for ChestXray dataset with ChexPert as the test domain, where the generalization gap is higher but the MI attack accuracy is low. We thus present our first result,

> Result 1: Out-of-Distribution generalization error cannot always explain the membership inference risk of a model.

## 4.2 STABLE FEATURES AND PRIVACY

Our second observation is that an algorithm's ability to learn stable features is a consistent indicator of its robustness to membership privacy. Since the true stable features are unknown in real data, our results are based on a proxy metric.

To verify the results, we use the slab dataset introduced for simplicity bias detection from Shah et al. (2020), with the modifications for the domain generalization setup as suggested by Mahajan et al. (2021). Compared to MNIST and ChestXrays datasets, the slab dataset provides us with fine-grained control to modify any specific feature and measure stable features accurately.

**Dataset.** This dataset has two features for a binary classification task, where the first feature (linear feature) has a spurious linear relationship with the label, while the second feature (slab feature) has a stable piece-wise linear relationship with the label. The relationship between the linear feature and the label is made spurious by adding domain dependent noise. We train the models on two source domains, with the probability of corruption in the linear feature–label mechanism as p=0.0 and p=0.1 in respective domains. For the test data, we consider two different unseen target domains, where one corresponds to the case of small distribution shift (p=0.2) and large distribution shift (p=0.9) in

the linear feature–label prediction mechanism. Note there is a constant domain independent noise (p=0.1) in the slab feature–label relationship, which makes it invariant across domains. Further implementation details are provided in Supp. B.2.

The Linear-Randomization AUC metric (Shah et al., 2020) measures the AUC when the linear feature is randomized. It captures the extent to which a model relies on the spurious linear features for prediction (and consequently, a measure for stable features; further details in Supp. A.4). The other metrics, OOD accuracy and Loss-based MI attack are computed as in Section 3. Since we know the stable slab features already, we also use an Oracle method on this dataset where the spurious linear features are masked out with a constant value during training, which ensures that it would only learn stable features.

**Results:** Figure 5 shows the results for the two test datasets with low and high distribution shift respectively. A higher linear-RAUC value indicates that a model has learnt more stable features. Unlike the OOD accuracy, the Linear-RAUC values has a consistent inverse correlation with MI attack accuracy (see especially Test Domain 0.2 with small distribution shift). With an accurate measure for stable features, this experiment confirms that robustness against MI attacks is correlated with stable features, under both small and high distribution shifts.

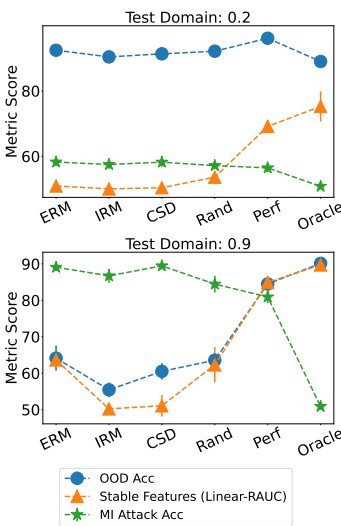

Figure 5: Results for synthetic dataset where Linear-AUC measures the learnt stable features and explains the membership attack accuracy of the models.

> Result 2: Models that learn stable features are empirically more robust to MI attacks than others.

One way to interpret the connection of stable features to privacy is to decompose the probability of model's loss, which is what is compared by any loss-based membership inference attack.

$$p(l(y, \hat{y})) = \sum_{\mathbf{x}} p(l(y, \hat{y})|\mathbf{x})p(\mathbf{x}) \qquad (1)$$

When the distribution of input $\mathbf{x}$ is changed, both $p(x)$ and $p(l(y, \hat{y})|x)$ can change, since the loss may vary for different inputs. However, if a model captures stable features $\mathbf{x}_c$, then its loss distribution will stay the same for all inputs since true function $y = f(x_c)$ remains the same (and models capturing a higher degree of stable features are expected to be more stable in their loss). This stability in loss helps such models obtain a lower change to the $p(l(y, \hat{y}))$ that is measured by membership inference, for the same change in input features.

### 4.3 MI ATTACK AS A METRIC FOR DG

Another interesting point to note is that in most of the current DG work the OOD accuracy is used for model selection. However, the relationship between OOD accuracy and stable features is sensitive to the choice of the target domain. Figure 5 (left) represents two target domains, where the target domain with slab noise 0.2 is more similar to the source domains as compared to the case of target domain with slab noise 0.9 (Figure 5 (right)). When the target domain is similar to the source domains, we expect the models relying on spurious features to obtain good OOD accuracy as well. This is shown in Figure 5 (left) where methods like ERM, Random-Match that do not capture stable features (low Linear Randomized AUC score) also achieve good OOD accuracy as compared to methods like Perfect Match and Oracle that capture stable features. However, in this case OOD accuracy misleads us about the stable features, as the OOD accuracy of Oracle is smaller than ERM. When the target domain has higher distribution shift, only methods relying on stable feature are able to achieve good OOD accuracy. Therefore, we find that OOD accuracy reflects stable features only when the target domain has an extreme distribution shift. However, MI attacks are correlated with stable features (high Linear Randomized AUC) in both the cases. Therefore, MI attacks are a viable complement to OOD accuracy for model selection in DG and will be especially useful when domains do not exhibit extreme distribution shift.

## 5 ATTRIBUTE INFERENCE (AI) ATTACKS

Attribute inference attacks aim to infer the value of a specific (sensitive) attribute of the training datapoints from the model (Melis et al., 2019; Ganju et al., 2018; Ateniese et al., 2015). However, if the feature is sensitive but spurious i.e., not necessarily required for a correct prediction, ideally a model should not learn and hence leak it. Simply ignoring that attribute during training is not enough (Zhang et al., 2020; Song & Shmatikov, 2020). To mitigate AI attacks, a model's output should be statistically independent of the attribute's value which aligns with the goal in the domain generalization tasks. This motivates us to evaluate the performance of DG methods for AI attacks.

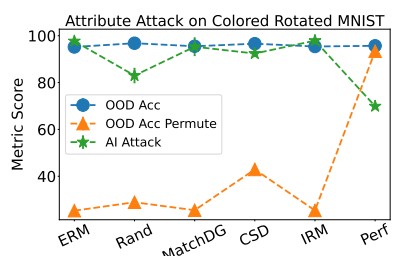

Figure 6: AI Attack Accuracy for Rotated-MNIST with color as an attribute.

As with MI attacks, we observe a similar benefit on attribute inference attacks for models that learn stable features such as Perfect-Match as shown in Figure 6. This holds whenever the sensitive attribute is not a causal feature for the prediction task, and thus can be ignored by the stable learning algorithm. We provide detailed attack description and empirical results for domain-agnostic attribute (color) and domain (angle) as the sensitive attribute for Rotated Digits MNIST dataset in Supp. C.2.

## 6 RELATED WORK

We discuss other closely related work on domain generalization and membership inference attack evaluation that is not covered in Section 2.

**Other Domain Generalization Approaches:** Early work in domain generalization focused on proposing regularizers to the standard training objective of minimizing loss over all training domains (Motiian et al., 2017; Dou et al., 2019). For instance, a simple regularizer-based method is to ensure that same-class inputs from different domains have the same output representation from the model. This helps to align inputs from different domains, but does not learn stable features since inputs from the same class can differ even within a single domain, and do not necessarily share the same stable (causal) features (Mahajan et al., 2021). Other methods (Magliacane et al., 2018; Gong et al., 2016; Muandet et al., 2013; Li et al., 2018) have proposed learning a representation $\Phi(x)$ such that the distribution of the representation learnt $P(\Phi(x)$ or $P(\Phi(x)|Y)$ remains the same across domains, but they suffer from similar issues on being reliant on the class label. In this work, we focus on DG algorithms that are designed to capture stable features as discussed in Section 2.

**Empirical study of Membership Inference (MI) Attacks:** Several prior work have performed empirical studies trying to understand MI attacks with different goals. Jayaraman & Evans (2019) and Rahman et al. (2018) demonstrate the efficacy of membership inference attacks when models are trained using differential privacy as a defense. Their main goal is to understand how different $\epsilon$ values for differential privacy affect membership inference attack accuracy. Song & Mittal (2020) perform a systematic evaluation using different method for MI attacks and demonstrate the efficacy of defenses such as adversarial regularization (Nasr et al., 2018) and Memguard (Jia et al., 2019). More recently, Francis et al. (2021) presented a similar study on evaluating membership and attribute inference for domain generalization method but in the federated learning setup. In this work, we aim to evaluate their effectiveness on domain generalization algorithms as well as understand their connection to stable features and out-of-distribution generalization.

## 7 CONCLUSION

We present an extensive empirical study on understanding the connection between out-of-distribution generalization, membership privacy and stable features using state-of-the-art domain generalization methods. Our results provide the first empirical evidence that OOD generalization gap alone cannot explain the risk of membership inference attacks of a given model i.e., a model with higher generalization gap may be more robust to MI attacks than one with lower generalization gap. From our experiments, we claim that the model's ability to capture stable features is what guarantees its robustness to membership privacy.

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

# The Connection between Out-of-Distribution Generalization and Privacy of ML Models

## Supplementary Material

## A  EVALUATION METRICS

We evaluate the privacy and generalization properties of DG training methods using four different metrics: membership inference accuracy, attribute inference accuracy, out-of-domain task accuracy, and the ability to learn stable features.

### A.1  MEMBERSHIP INFERENCE (MI) ATTACKS

Several methods have been proposed in the literature to compute MI attack accuracy based on the threat-model i.e., black-box or white-box and computational power of the attacker. All these methods identify the boundary that helps to distinguish between members and non-members. As our goal is to use MI attack accuracy as a measure for privacy and perform a comparison across several method, we use the loss-based attack to measure privacy.

**Loss-based attack Yeom et al. (2018)**  Yeom et al. (2018) proposed MI attack that relies on the loss of the target model. The attacker observes the loss values of a few samples and identifies a threshold that distinguishes members from non-members. The intuition is that training data points will have a lower loss value as compared to test data points. This attack is computationally cheap and the attacker does not need to train shadow models or an attack classifier. However, the attack assumes access to the loss values of the target model i.e., it requires white-box access to the model. Our attack accuracy provides an upper bound as we compute the threshold using a subset of the training samples and is consistent across all our experiments.

### A.2  ATTRIBUTE (PROPERTY) INFERENCE (AI) ATTACKS

We use a classifier-based attribute inference attack as our second metric to evaluate the privacy of different training techniques that aim for domain generalization. The attack is similar to the MI attack except that we build a classifier to learn the distinguishing boundary. We query a subset of data points to the target model and use their prediction probability vector as input feature for the attack classifier. The ground truth is the value of the sensitive attribute for the given input. The classifier is trained to predict the attribute value for a given input feature of probability vector. This attack works in a black-box setting.

### A.3  OUT-OF-DOMAIN ACCURACY

To understand the generalization ability of a training technique, we use the accuracy as a measure when computed on inputs that are generated from domains that are *not* seen during training. This is different than standard test accuracy measure where often the validation and test data have the same distribution. Since our goal is to understand the connection between domain generalization techniques and privacy, we select out-of-domain accuracy as one of our evaluation metrics.

### A.4  MEASURING STABLE FEATURES USING MEAN RANK

Measuring stability of learnt features is a hard task, since the ground-truth stable features are unknown for a given classification task. For example, in a MNIST task to classify the digit corresponding to an input image, the shape of the digit can be considered as the stable (or causal) feature whereeas its color or rotation are not stable features. Even if stable features are known, in image datasets, they are typically high-level features (such as shape) that themselves need to be learnt from data. Therefore, verifying stable features directly is a non-trivial task. We describe two different ways to measure stable features — mean rank and linear-Randomized AUC for our MNIST and synthetic slab dataset results below. However, we do not have a reasonable metric to measure stable features for ChestXray dataset and leave that to future work.

**Mean Rank.** We use the mean rank metric proposed by  Mahajan et al. (2021) for measuring stable features where the base object of an image is known. If we can select pairs of inputs with the same

base object, then they should share the same causal features. Such a pair is known as a perfect match. Note that a base object refers to the same semantic input such as a person or a handwritten digit. Input images may consist of the same person in different views or the same handwritten digit in different colors or rotations, but their base object remains the same. Note that there is a many-to-one relationship between an object and its class label. Each class label consists of many objects, which in turn consist of many input images that are differentiated by certain non-stable features like view or rotation or noise.

Formally, the mean rank metrics is computed as follows. For the matches (j, k) as per the ground-truth perfect match strategy $\Omega$, compute the mean rank for the data point j w.r.t the learnt match strategy $\Omega'$ i.e. $S_{\Omega'}(j)$

$$\frac{\sum_{\Omega(j,k)=1;d\neq d'} Rank[k \in S_{\Omega'}(j)]}{\sum_{\Omega(j,k)=1;d\neq d'} 1} \tag{2}$$

**Linear-Randomized AUC.** Slab dataset allows to capture the effect of a single feature on the outcome prediction by constructing S-Randomized metrics (Shah et al., 2020). The idea is to replace a subset of features by drawing random samples from its marginal distribution so that we destroy any relationship between the features in the set S and the outcome y. As defined in the work (Shah et al., 2020), consider the data point $x = (x^S, x^{S^c})$, and replace the features $x^S$ in the actual dataset $((x^S, x^{S^c}), y) \sim \mathcal{D}$ with new samples $\bar{x}^S \sim \mathcal{D}_S$, where $\mathcal{D}_S$ is the marginal distribution of features in the subset S. Denote this new dataset as S-Randomized and compute metrics like accuracy and AUC on it. Since the relationship between the features in set S and the outcome y has been randomized, the S-Randomized AUC/Accuracy would be close to 0.50 if the model completely relied on the features in the set S for prediction.

Hence, we take the set S to be the spurious linear feature and compute the Linear-Randomized AUC score, which captures the extent to which a model relies on the spurious feature for its prediction. Therefore, models that capture more stable features would have higher values of Linear-Randmoized AUC, as the randomization in the prediction mechanism between the linear feature and the label would affect their performance comparatively less.

## B  EXPERIMENTAL SETUP

### B.1  OOD TRAINING METHODS AND ERM

For all methods and the respective loss equations, we use $S$ to denote the set of source domains, $N_d$ as the total number of samples for domain $d$, $f$ as the classification model, $L_d$ as the classification loss, and $x, y$ to represent the data point and its corresponding true class label.

**ERM-Baseline:**  As our baseline, we use the empirical risk minimization approach to train the model, which minimizes the empirical average of loss over training data points. $\sum_{d\sim S, i\sim N_d} L_d(f(x_i), y_i)$

It treats the data from different domains as i.i.d and simply augments them. This may lead to issues with OOD generalization Arjovsky et al. (2019); Peters et al. (2016) as we need to learn representations that are robust to the changes in the domains. Hence, a variety of approaches (described below) augment the empirical average loss with regularizers to learn domain invariant models.

**Random-Match (Mahajan et al., 2021; Motiian et al., 2017; Dou et al., 2019).**  Random-Match matches pairs of same-class data points randomly across domains to regularize the model. The idea behind matching across domains is to learn a representations that is invariant to the changes in the source domains, which may lead to better generalization performance. The training loss objective is given by,

$$\sum_{d\sim S, i\sim N_d} L_d(h \circ \phi(x_i), y_i) + \lambda * \sum_{\Omega(j,k)=1|j\sim N_d, k\sim N_{d'}} Dist(\phi(x_j), \phi(x_k)) \tag{3}$$

where $\phi$ represents some layer of the network $f = h \circ \phi$, $\Omega$ represents the match function used to randomly pair the data points across the different domains. This may not necessarily enforce learning stable features.

**CSD (Piratla et al., 2020).** Common-Specific Low-Rank Decomposition (CSD) leads to effective OOD generalization by separating the domain specific and domain invariant parameters, and utilizes the domain invariant parameters for reliable prediction on OOD data. It decomposes the model's final classification layer parameters $w$ as $w = w_s + W * \gamma$, where $W$ represents the k-rank decomposition matrix, $w_s$ represents the domain invariant parameters and $\gamma$ represent the k domain specific parameters. It optimizes empirical average loss with both the domain invariant and domain specific parameters, along with an orthonormality regularizer that aims to make $w_s$ orthogonal to the decomposition matrix $W$. Please refer to Algorithm 1 in their paper Piratla et al. (2020) for more details.

**IRM (Arjovsky et al., 2019).** Invariant Risk Minimization (IRM) aims to learn invariant predictors that simultaneously achieve optimal empirical risk on all the data domains. It minimizes the empirical average loss, and regularizes the model by the norm of gradient of the loss at each source domain as follows:

$$\sum_{d \sim S, i \sim N_d} L_d(w \circ \phi(x_i), y_i) + \lambda * \sum_{d \sim S} ||\nabla_{w|w=1.0} \sum_{i \sim N_d} L_d(w \circ \phi(x_i), y_i)||^2 \tag{4}$$

where $f = w \circ \phi$ and $\lambda$ is a hyper parameter. In practice, $\phi$ is taken to be the final layer of the model $f$, (which makes $\phi$ and $f$ to be the same ). Hence, minimizing the above loss would lead to low norm of the domain specific loss function's gradient and guide the model towards learning an invariant classifier, which is optimal for all the source domains.

**MatchDG (Mahajan et al., 2021).** The algorithm enforces the same representation for pairs of data points from different domains that share the same causal features. It uses contrastive learning to learn a *matching* function to obtain pairs that share stable causal features between them. The loss function for the method is similar to that of Random-Match (Eq 3), with $\Omega$ representing the match function learnt by contrastive loss minimization. Hence, the algorithm consists of two phases; where it learns the matching function $\Omega$ in the first phase, and then minimizes the loss function in Eq 3 during the second phase to learn the final model. Please refer to the Algorithm 1 in Mahajan et al. Mahajan et al. (2021) for more details.

**Perfect-Match (Mahajan et al., 2021; Hendrycks et al., 2019).** Finally, we use an algorithm that can be considered to learn *true stable* features for given data, since it relies on knowledge of true base object for a subset of images (and thus guaranteed shared causal features between them). This approach again has a similar formulation to Eq 3, where the match function $\Omega$ is satisfied for data points from different domains that share the same base causal object. Hence, it aims to learn similar representations for two data points that only differ in terms of the domain specific attributes.

**Hybrid (Mahajan et al., 2021).** Perfect matches as explained above are often unobserved but given through Oracle access. However, in real datasets, augmentations can also provide perfect matches, leading to the *Hybrid* approach using both the MatchDG and augmented Perfect-Match. It learns two match functions, one on the different source domains as per MatchDG, and the other on the augmented domains using Perfect-Match.

Note that Perfect-Match is an ideal training algorithm that assumes knowledge of ground-truth matches across domains, and therefore cannot be applied in real-world settings. In contrast, the Hybrid algorithm depends on creating matches of the same base object using self-augmentations and can be used practically whenever augmentations are easy to create (such as in image datasets). In the experiments that follow, we use the PerfectMatch algorithm for the simulated Rotated-MNIST and Fashion-MNIST datasets, where it should be considered as an ideal method. For the real-world Chest X-rays dataset, we use the practical Hybrid algorithm since we have no knowledge about the true perfect matches.

| Dataset | #Classes | #Domains | Source Domains | Target Domains | Samples/ Domain |
|---------|----------|----------|----------------|----------------|-----------------|
| Rotated-MNIST | 10 | 7 | $15°, 30°, 45°, 60°, 75°$ | $0°, 90°$ | 2000 |
| Fashion-MNIST | 10 | 7 | $15°, 30°, 45°, 60°, 75°$ | $0°, 90°$ | 2000 |
| ChestXray | 7 | 3 | NIH, ChexPert | RSNA | 800 |

Table 1: Dataset details

Table 2: Hyperparamter details for all the datasets. We took the optimal hyperparameter for each method following Mahajan et al. (2021). Complete details regarding the grid range used for hyperparameter tuning can be found in Table 8 in their paper.

| Dataset Range | Hyper Parameter | Optimal Value |
|---------------|-----------------|---------------|
| Rotated & Fashion MNIST | Total Epochs | 25 |
| | Learning Rate | 0.01 |
| | Batch Size | 16 |
| | Weight Decay | 0.0005 |
| | Match Penalty | 0.1 |
| | IRM Penalty | 1.0 (RotMNIST); 0.05 (FashionMNIST) |
| | IRM Threshold | 5 (RotMNIST), 0 (FashionMNIST) |
| Chest X-ray | Total Epochs | 40 |
| | Learning Rate | 0.001 |
| | Batch Size | 16 |
| | Weight Decay | 0.0005 |
| | Match Penalty | 10.0 (`RandMatch`), 50.0 (`MatchDG`, `MDGHybrid`) |
| | IRM Penalty | 10.0 |
| | IRM Threshold | 5 |
| Slab Dataset | Total Epochs | 100 |
| | Learning Rate | 0.1 |
| | Batch Size | 128 |
| | Weight Decay | 0.0005 |
| | Match Penalty | 1.0 |
| | IRM Penalty | 10.0 |
| | IRM Threshold | 2 |

## B.2 IMPLEMENTATION DETAILS

**Model Training** To summarize, we use the model ResNet-18 (no pre-training) for the Rotated-MNIST and Fashion-MNIST dataset, and we use pre -trained DenseNet-121 for the ChestXRay dataset. For the matching based methods (Random-Match, MatchDG, Perfect-Match), we use the final classification layer of the network as $\phi$ and for the matching loss regularizer (Eq 3 ). For all the methods across datasets, we use Cross Entropy for the classification loss ($L_d$), and use SGD to optimize the loss. Also, we use the data from the source domains for validation and never expose the model to any data from the target domains while training. The details regarding the domain and dataset sizes are described in the Table 1.

For the slab dataset, we sample 1k data points per domain and additional 250 data points per source domain for validation. We construct two source domains, with noise probabilities between the linear feature and the label as p=0.0 and p=0.1, while there is constant domain-independent noise (p=0.1) in prediction mechanism between the slab feature and the label. The model architecture consists of a representation layer (FC: input-dimension * 100; ReLU activation ), followed by a classification layer ( FC: 100*100; FC: 100*total-classes), where FC: a*b denotes a full connected layer with input and output dimension as a, b respectively. For the matching based methods (Random-Match, MatchDG, Perfect-Match), the representation layer is taken as $\phi$.

SGD is used to optimize the training loss across all the datasets and methods, with the details regarding other hyperparameters given in the table 2.

**Membership Inference (MI) Attacks.** For implementing MI attacks, we first create the attack-train and attack-test dataset from the original train and test dataset for the ML model. We first sample $N$ number of data points ( N: $2,00$ for Rotated-MNIST & Fashion-MNIST, N: $1,000$ for ChestXRay, N :$400$ for Slab dataset ) from both the original train and test dataset to create the attack-train dataset. Similarly, we sample an additional set of $N$ data points from original train, test dataset to create the attack-test dataset. For the selection of the threshold $\tau$ required for the Loss-based attack, we compute the mean loss among all the members in the attack-train dataset. We use this threshold to evaluate performance on the attack-test dataset.

**Attribute Inference (AI) Attacks.** We use all the source and target domains, and sample data points from their training/test set to create the attack-train/attack-test dataset respectively. We then label data points in the attack-train and attack-test dataset based on the attribute, and then train a classifier to discriminate among different attributes. We train a 2-layer fully-connected network ( with hidden dimensions 8, 4 respectively ) to distinguish between different attributes for all the models. We use Adam Optimizer with learning rate 0.001, batch size 64 and 5000 steps / 80 epochs.

**DP Training.** We train the differentially-private models using Pytorch Opacus library support for DP-SGD training. We train models with $\epsilon$ ranging from 1 to 10. We use max clipping norm of 5.0 and the noise multiplier is inferred from the $\epsilon$ value. The other hyperparamters like total number of epochs, learning rate, etc. are same for the non DP case in Rotated-MNIST & Fashion-MNIST, with the exception of a higher batch size ($10 \times$ the usual batch size) as it typically helps with dp training.

## C  ADDITIONAL EXPERIMENTAL RESULTS

### C.1  SLAB DATASET

Figure 7 shows the result for our synthetic slab dataset with an additional metric of train-test generalization gap. We find that for this dataset, the generalization gap metric can be used to measure the amount of stable features learnt which in turn correlates with MI attack accuracy for different training algorithms.

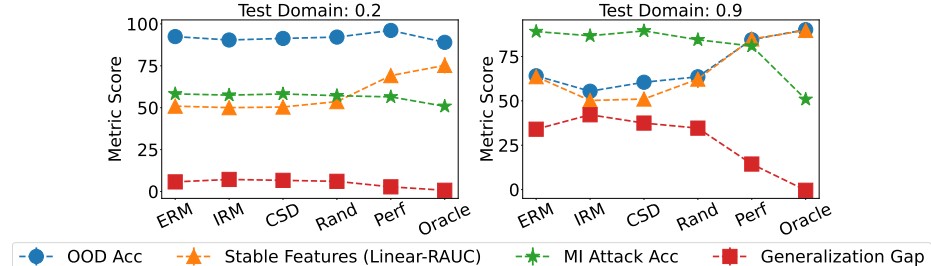

Figure 7: Results for synthetic slab dataset with generalization gap included.

Table 3: Attribute Attack Accuracy for the dataset Rotated-MNIST with color as an attribute. We consider the attribute as binary (0: no color; 1: color ), thus the best case AI Attack accuracy would be 50% via random guess.

| Metrics | ERM | Random-Match | IRM | CSD | MatchDG | Perfect-Match/ Hybrid |
|---|---|---|---|---|---|---|
| Train Accuracy | 99.7 (0.02) | 99.8 (0.05) | 99.7 (0.02) | 99.9 (0.01) | 99.4 (0.10) | 97.8 (0.31) |
| OOD Accuracy | 95.2 (0.17) | **96.8 (0.06)** | 95.4 (0.21) | 96.6 (0.48) | 95.5 (0.28) | 95.7 (0.19) |
| AI Attack Accuracy | 97.7 (0.41) | 83.0 (3.09) | 97.9 (0.30) | 92.4 (1.30) | 95.3 (3.80) | **69.9 (0.03)** |
| OOD (Permute) Accuracy | 25.3 (0.17) | 28.9 (1.81) | 25.4 (0.21) | 42.9 (2.30) | 25.5 (0.28) | **93.4 (0.35)** |

Table 4: Attribute Attack Accuracy for different datasets with domains as an attribute.

| Dataset | Random Guess | ERM | Random-Match | IRM | CSD | MatchDG | Perfect-Match/ Hybrid |
|---|---|---|---|---|---|---|---|
| Rotated-MNIST | 14.3% | 27.6 (0.84) | 20.4 (0.46) | 26.6 (0.71) | 25.0 (0.40) | 19.3 (0.62) | **16.6 (0.55)** |
| Fashion-MNIST | 14.3% | 26.6 (0.59) | 21.8 (0.77) | 23.8 (0.86) | 26.9 (0.93) | 21.9 (0.29) | **21.5 (0.97)** |
| ChestXray | 33.3% | 61.8 (1.02) | 58.4 (0.58) | 63.4 (2.28) | 67.8 (3.05) | 57.9 (0.58) | **57.1 (0.21)** |

## C.2 ATTRIBUTE ATTACKS

**Domain-Agnostic Attribute:** We consider a domain-agnostic attribute — color in the rotated digit dataset which is not related to the prediction task and whose distribution does not depend on the domain. We randomly introduce color as an attribute with 70% probability to Rotated-MNIST dataset. The attacker's goal is to infer whether a given input is colored or not (binary task) with only access to the output prediction. This setting can be observed for example, where the user sends embedding of their input to the server instead of the original input (Song & Shmatikov, 2020).

Figure 6 shows our main results with additional results in Table 3. We observe that Perfect-Match has the lowest attack accuracy as compared to all other training algorithms, demonstrating that the ability to learn stable features provides inherent ability to defend against a harder task of attribute inference when the attribute itself is domain-agnostic. Note that the OOD accuracy does not convey much information about the privacy risk through an AI attack: all methods obtain OOD accuracy within a narrow range of 95.9%-97.5%, but the attack accuracy range from 70.2% for Perfect-Match to 99.3% for IRM.

We hypothesize that the difference in attack accuracy is because of the differing extents to which the ML models utilize color as a feature. To verify the hypothesis, we construct a new, *permuted* test domain where the color of each colored image is permuted randomly to a different value. Thus, the correlation in the training data between color and the class label is no longer present in the test domain. Under such a test domain, we observe larger differences in OOD accuracy that correspond to the AI attack accuracy reported above: Perfect-Match obtains the highest OOD accuracy (94.9%) as compared to other methods that range between 28% to 42%. The low accuracy of other DG methods motivates further research to introduce constraints for learning stable features.

**Domain as a sensitive attribute:** In this attack, we consider domain of the data from which it is generated as a sensitive attribute that the attacker is trying to learn. Table 4 shows the attack results across different training methods on all the three datasets. For Rotated-MNIST and Fashion-MNIST, we train the attack classifier to distinguish among all the 7 angles of rotation and hence the baseline accuracy with random guess 14.33% while for ChestXray, we aim to distinguish among 3 domains resulting in a baseline of 33.33%.

Similar to MI attacks, we observe that PerfectMatch/Hybrid approach has the lowest attack accuracy followed by MatchDG. PerfectMatch/Hybrid, having access to perfect matches based on self-augmentations, has access to the pairs of inputs that share the same causal features, and hence obtains both highest OOD accuracy (as we saw earlier) and the lowest AI attack accuracy. However, there is not a one-to-one correspondence between OOD accuracy and AI attack accuracy. CSD obtains one of the highest OOD accuracies on all datasets but its attack accuracy is one of the highest. We suspect that this discrepancy is due to the different objective of CSD algorithm compared to the matching algorithms of PerfectMatch/Hybrid and MatchDG: CSD does not aim to obtain the same representations across domains. Thus, when domains convey sensitive information, it is preferable to employ matching-based methods that can provide both high OOD accuracy and low attribute inference attack accuracy. Unlike in membership inference attacks, the standard ERM training algorithm does not perform well on attribute inference, obtaining one of the highest attack accuracies.

That said, it is somewhat intuitive that DG training approaches that are designed to be domain invariant are able to defend against attribute inference attacks as compared to ERM. Therefore, we present another attack with a domain-agnostic attribute to understand whether DG methods can mitigate such attacks well.

