# OpenReview forum: "The Connection between Out-of-Distribution Generalization and Privacy of ML Models"
_ICLR.cc/2022/Conference — ICLR 2022 Submitted_

### Official Review · Reviewer_JCc9 · 2021-10-31

**Correctness:** 2
**Technical Novelty And Significance:** 2
**Empirical Novelty And Significance:** 3
**Recommendation:** 3
**Confidence:** 4

**Main Review:**

Strengths:
+ OOD generalization is important and hard to measure without labeled data from out-domain distribution. The MI membership attack depends on the availability of unlabeled data from all domains and the prediction loss of the model in distinguishing different domains is used as the indicator.
+ State-of-the-art OOD methods are evaluated and there are some consistent patterns in the correlation between MI attack accuracy and OOD accuracy. This is done on multiple datasets.
+ Include many scenarios, such as the attribute inference attack and stable feature learning.

Weaknesses:
- In some cases, the correlation is not very clear or significant. For example, Figure 3 shows that MatchDG and Hybrid have similar MI attack accuracy (right panel), but their OOD accuracies are very different (left panel). In the same figure, it is not clear what the right panel is showing (is "Train-test accuracy gap" the same as the OOD gap?)
- The paper is overly an empirical paper. There is no theory. What's more, there is no explanation why the correlations hold true in some cases. The "explanations" are mostly about the observation but not conjectures about why the observations are so. This weakness leads to limited insight from the paper.
- The authors try to contrast the OOD method and privacy-preservation method (DP). There is unfair comparison, however. For example, the in-domain generalization accuracy is not shown. Will DP with a reasonable $\epsilon$ have both high in-domain generalization accuracy and low MI attack accuracy?
- The correlation between OOD generalization and MI attack accuracy is less obvious on the real-world datasets in Figure 3. This is important since there can be other factors that determine OOD accuracy beyond the MI attack accuracy. The authors should consider explaining some of such possible factors.
- paper organization: sections 4.2 and 4.3 seem different but overlapping section 3. Section 4 contains more empirical results to explain the empirical results in section 3. However, as mentioned above, some in-depth insight or theory will better explain the results in Section 3. If the findings are different from the prior work Yeom et al. (2018) and Tople et al. (2020), then what are the possible reasons?

**Summary Of The Paper:**

The paper proposes to use the metric called "membership inference (MI) attack accuracy" as an indicator to determine the OOD generalization accuracy. The benefit of using such a metric is that no labeled data Pr(Y, X) is needed from the test data but only the test data Pr(X). The paper studies the correlation between MI attack accuracy and OOD accuracy on synthetic and real-world datasets. There is no theoretical treatment of the relationship but prior work Yeom et al. (2018) and Tople et al. (2020) was cited as the motivation.

**Summary Of The Review:**

The paper is well motivated. However, the empirical studies have mixed results (especially on the real-world datasets) while the explanations of the observerations are neither not in-depth or comprehensive.

---

> ### Author Response · Authors · 2021-11-23
> **Clarification of Results**
>
> 1. In Figure 3, the OOD accuracy is simply the accuracy of the model on the test dataset and Train-test gap is the OOD generalization gap. Figure 3 shows that the MI attack accuracy for Match DG is always slightly higher for RSNA and NIH than Hybrid and it is the same for the train-test accuracy gap as well.  For Chest-Xray dataset where stable features are not known, Figure 3 shows that OOD generalization gap is correlated to MI attack accuracy, with models that are known to capture stable features while training (such as MatchDG or Hybrid) provide the best privacy guarantee as compared to others. We find that the OOD accuracy has a similar trend (i.e., MatchDG has lower test accuracy than Hybrid) but with a larger difference in absolute values.
>
> 2. The theoretical connection between models that learn stable (causal) features and different privacy as well as MI attacks is established in prior work by Tople et al. under the assumptions that the true graph is known. Our focus is to analyse state-of-the-art models that are designed to learn stable features without any prior knowledge about the true stable features and empirically evaluate their membership privacy guarantees when trained on high dimensional input data such as images. Such purely empirical work is important for understanding to what extent do theoretical guarantees exhibit in practice in real-world settings.
>
> 3. We do not aim to contrast OOD methods and DP but our goal is to demonstrate that methods that learn stable features such as Perf-Match provide higher utility for the same epsilon-DP guarantees as compared to the baseline ERM method. With regards to the empirical privacy as measured via MI attack we observe that higher epsilon values such as 5 or 10 do not provide any additional protection against MI adversary as compared to a non-DP model. Variant of these results have been show in prior work for the in-distribution setting [Humphries et al.].
>
> Humphries et al. " Differentially Private Learning Does Not Bound Membership Inference" Arxiv
>
> 4. We will better organize the paper structure for clarity, thanks for your suggestion.
>
> 5. We iterate that our findings are not different than Yeom et al. and Tople et al. but are presented for different setting of Out-of-Distribution prediction tasks and in real-world scenarios where the assumptions such as knowledge of true structural causal model (SCM) are not always satisfied.

---

### Official Review · Reviewer_upDo · 2021-11-02

**Correctness:** 3
**Technical Novelty And Significance:** 3
**Empirical Novelty And Significance:** 2
**Recommendation:** 5
**Confidence:** 3

**Main Review:**

The paper addresses an important problem for the community since it is not easy to measure without labeled data from out-domain distribution. To my understanding, membership inference attacks depend on the availability of unlabeled data from all domains. The authors evaluated SOTA methods on various datasets and found some patterns in the correlation between MI attack accuracy and OOD accuracy.

However on some datasets, the correlation is not very clear. Given that no theoretical analysis are provided, it is difficult to come to a conclusion in these cases. The explanations only depends on empirical evaluation, but at least some theoretical support is also required for these explanations. In addition to that, it is difficult to follow the paper, it requires restructuring. For example, Section3 is divided for datasets and I'd expect to see the same subtitles for 3.1 and 3.2 but they are different. Another example is why 4.3 is part of section 4 and not a separate section like section 5?


**Summary Of The Paper:**

This paper presents an extensive empirical study to show the connection between out-of-distribution (OOD) generalization, privacy and stable features using SOTA domain generalization methods. They propose to use MI attack metric, which requires no labeled data Pr(Y, X) but only the test data Pr(X), to evaluate quality of the learnt features, because they find out that it measures stable features better than OOD accuracy. The results shows that there is no direct relationship between better OOD generalization and privacy. They empirically prove that a higher generalization gap is not necessary to perform membership inference (MI) attacks in OOD deployment scenarios. They also show that for the same added noise for DP, an algorithm with stable features provides better utility.

**Summary Of The Review:**

This paper makes an extensive evaluation to show the connection between out-of-distribution (OOD) generalization and privacy. It is important for privacy community, but I'm not sure if it is complete without any theoretical claim. Can the authors provide a theoretical analysis to show the connection and support their claims?

---

> ### Author Response · Authors · 2021-11-23
> **Theoretical and Empirical Results**
>
> 1. The theoretical connection between stable features and differential privacy (DP) as well as membership inference (MI) attack accuracy has been show in prior work by Tople et al. However, their work assumes the knowledge of true stable (causal) features. The main focus of our work is to analyse if such relation exists in practical settings on real-world data when models are trained with state-of-the-art learning algorithms and the underlying true causal features are unknown.
>
> 2. Thank you for the comments on restructuring the paper. We will incorporate them in the updated version of the paper.

---

### Official Review · Reviewer_eEMp · 2021-11-03

**Correctness:** 3
**Technical Novelty And Significance:** 2
**Empirical Novelty And Significance:** 3
**Recommendation:** 5
**Confidence:** 3

**Main Review:**

Strength:
This work studies an important relationship between OOD generalization, stable features, and privacy. The authors performed extensive experiments to study their connections. The empirical results also shows that robustness to MI attacks can be a good indicator of whether the learned features are stable.

Weakness:
The conclusions are drawn mainly by comparing different models/domain generalization algorithms. Therefore, there might be other unknown hidden factors that lead to different performance for these algorithms. I think the results would be more convincing if you can include experiments where you fix an algorithm and somehow change the OOD performance/stable features. Not sure if such control experiment is possible, but my very naïve guess is that if you corrupt part of the training dataset, the resulting feature might not be stable across different domains? If such control experiment cannot be done, please explain the main difficulties in designing such experiments.

**Summary Of The Paper:**

This works studies the relationship between membership privacy and out-of-distribution (OOD) generalization/stable features. Through extensive experiments, the author shows that the ability to capture stable features lead to robustness against membership inference attacks, while it is not necessarily the case for OOD generalization.

**Summary Of The Review:**

This work studies the connection between OOD generalization, stable features and privacy (particularly robustness against membership attacks) through extensive experiments. It is mainly an empirical study, so technical contribution is limited. However, it provides evidence on the relationship between the three factors. A particularly interesting insight is that robustness against MI attacks can be an indicator of stable features. The main concern is that the experiments do not control the factor of the algorithms being used.

---

> ### Author Response · Authors · 2021-11-23
> **Control experiments for Domain Generalization Algorithms**
>
> 1. The main problem is not in conducting such experiments but the lack of knowledge regarding the underlying stable features for the prediction task.  Your suggestion may be possible in simulated data like the Slab dataset. For real-world data,  our paper provides one set of ablation experiments where the number of training domains are reduced, thereby reducing the amount of information available to identify and hence learn stable features.
>
> 2. Based on your suggestion, a potential control experiment could be to reduce the percentage of ground truth matches present at initialization for the matching based methods as performed in the MatchDG paper (Table 13, Suppl.). This may be a good approximation for the required case as the underlying algorithm remains the same and the different initializations (Random Match to Perf Match) lead to varying amount of stable features captured. We will update the paper with these results.

---

### Official Review · Reviewer_GSgZ · 2021-11-10

**Correctness:** 2
**Technical Novelty And Significance:** 1
**Empirical Novelty And Significance:** 1
**Recommendation:** 1
**Confidence:** 4

**Main Review:**

The paper targets  an interesting question about the connection between generalization and privacy. However, there are some
Issues:

My main issue with the paper is the idea of using MI-attack-accuracy as a the definition of privacy risk. The membership attack metric only captures one type of adversary that is trying to learn the data. Depending on the implementation of the MI attack you might get very different accuracy and thus I don’t think it is a good metric for privacy in general. In my view differential privacy is a more principled privacy definition because it protects privacy agains the worst case adversary.

Also I was confused about the key results: I don’t find it surprising that achieving generalization does not provide more privacy and showing that there is not connection between generalization and privacy does not seem difficult to do. As for stable features, I don’t see a clear connection between stable features and differential privacy. For example  I would be interested in seeing is Mean-Rank vs Epsilon , to see if more differentiable privacy methods learn better stable features.
Other than that, I find the presentation of key results not very clear. For example in Figure 1 it’s not easy to see the correlated between OOD accuracy and MI attack accuracy. Would it make more sense to have a plot with OOD accuracy in the y-axis and MI-loss in the x-axis?


**Summary Of The Paper:**

The paper evaluates the connection between out of distribution (OOD) accuracy of a learning model and its privacy protection. They use Membership inference (MI) success rate as a metric to evaluate how much privacy some model provides.

The setting they evaluate is when a model is train data and test data are drawn from different distributions. The main idea is that models trained on data from some distribution should be robust to distribution shift. The key insight from the literature is that models that generalize to out-of-distribution data are able to learn stable features of the data (features that are invariant to modest distribution shifts). The paper empirically evaluates the connection between the generalization ability of a model and its privacy (using the MI metric).


**Summary Of The Review:**

The paper uses a questionable notion of privacy. Also, key results and presentation of experiments is not clear.

---

> ### Author Response · Authors · 2021-11-23
> **Membership Inference Attacks and Privacy Risk**
>
> 1. Our justification for using Membership inference attack as a measure for privacy risk is as follows:
> Theoretically, it has been shown that DP epsilon provides an upper bound to the MI attack accuracy of the adversary [Yeom et al. [CSF'18], Humphries et al].  Empirically, recent works  have shown that a DP adversary can be instantiated using membership inference attacks [Jagielski et al.(NeurIPS'20) , Nasr et al. (S&P'20)] which can be used to compute lower bound estimates of epsilon. As computing empirical estimates of epsilon lower bounds of a given model is computationally expensive (training thousands of models), we use MI attack as a proxy to measure the privacy risk.
>
> - Jagielski et al. "Auditing Differentially Private Machine Learning: How Private is Private SGD?", NeurIPS'20
> - Nasr et al. "Adversary Instantiation: Lower Bounds for Differentially Private Machine Learning", S&P'210
> - Humphries et al. " Differentially Private Learning Does Not Bound Membership Inference" Arxiv
> - Yeom et al. "Privacy Risk in Machine Learning: Analyzing the Connection to Overfitting", CSF'18
>
> 2. Based on the above argument, we chose to demonstrate connection between stable features and MI attack. We agree that mean-rank vs epsilon is another approach to demonstrate the same. We will implement this in the next version and add further experiments to show the connection between stable features and the amount of DP noise required to achieve similar epsilon guarantees.
>
> 3. We wanted to compare different models for both OOD accuracy and MI attack. We can provide OOD accuracy vs MI attack as suggested for a selected few to highlight the relationship better.

---

### Author Response · Authors · 2021-11-23
**Summary of Response**

We thank all the reviewers for their helpful feedback and appreciate the suggestions for improving the paper. We provide clarification responses  for each reviewer separately in the comments. We clarify our main contribution below:

Prior work has shown theoretical connection between generalization gap, stable features, differential privacy as well as membership inference attacks [Yeom et al., Tople et al.]. However, it is not clear if these results exhibit in practical and realistic setting where certain assumptions such as prior knowledge of true causal features or where train and test dataset correspond to the same distribution. In this work, we take the first step in empirically evaluated the membership privacy guarantees of both causal and non-causal models in the domain generalization setup where the causal or stable features are not known such as high dimensional image datasets.

Our results show that models that capture higher stable features often provide better robustness to membership inference attacks.

For the domain generalization community, these results are useful as MI attack metric can be used to evaluate quality of the learnt features.

---

### Decision · Program_Chairs · 2022-01-20

**Decision:**

Reject

**Comment:**

Motivated by the connections between privacy and generalization, this paper studies the correlation between MI attack accuracy and OOD accuracy on synthetic and real-world datasets. It shows that the measurements are not always correlated. I found the connection between the motivation and actual measurements performed in the experiments to be rather tenuous. Therefore it is hard to draw any insightful conclusions from the empirical results. It should also be noted that somewhat related disconnect between accuracy of MIA and generalization has already been observed in prior work.